# 🫘 PINTO: Faithful Language Reasoning Using Prompt-Generated Rationales

**Peifeng Wang**[1,2]**, Aaron Chan**[1]**, Filip Ilievski**[1,2]**, Muhao Chen**[1,2]**, Xiang Ren**[1,2]

[1]Department of Computer Science, University of Southern California
[2]Information Sciences Institute, University of Southern California
{peifengw, chanaaro, muhaoche, xiangren}@usc.edu, ilievski@isi.edu

## Abstract

Neural language models (LMs) have achieved impressive results on various language-based reasoning tasks by utilizing latent knowledge encoded in their own pretrained parameters. To make this reasoning process more explicit, recent works retrieve a rationalizing LM's internal knowledge by training/prompting it to generate free-text rationales, which can be used to guide task predictions made by either the same LM or a separate reasoning LM. However, rationalizing LMs require expensive rationale annotation, without any assurance that the generated rationales improve LM task performance or faithfully reflect LM decision-making. In this paper, we propose PINTO, an LM pipeline that *rationalizes* via prompt-based learning, and learns to *faithfully reason over rationales* via counterfactual regularization. First, PINTO maps out a suitable reasoning process for the task input by prompting a frozen rationalizing LM to generate a free-text rationale. Second, PINTO's reasoning LM is fine-tuned to solve the task using the generated rationale as context, while regularized to output less confident predictions when the rationale is perturbed. Across four datasets, we show that PINTO significantly improves the generalization ability of the reasoning LM, yielding higher performance on both in-distribution and out-of-distribution test sets. Also, PINTO leverages the rationales more faithfully than competitive baselines do.

## 1 Introduction

Many language-based reasoning tasks require retrieving and reasoning over knowledge beyond the task input—*e.g.,* commonsense reasoning and closed-book QA (Fig. 1, left) [29, 20]. Neural language models (LMs) have achieved impressive results on such tasks by utilizing latent knowledge encoded in their pretrained parameters [23, 4]. Still, given LMs' black-box nature, it is unclear whether this knowledge is being used properly [8, 16]. Previous studies have shown that LMs often learn spurious correlations from artifacts in downstream training data, thus limiting their generalizability [3, 9, 6].

With this in mind, a number of prior works aim to make LMs' reasoning processes more *explicit* by generating free-text rationales, which use LMs' internal knowledge to describe a reasoning process in natural language [21, 32, 18, 35]. In the *fine-tuned self-rationalizing* paradigm, a single LM is fine-tuned to jointly generate the task output and rationale [21, 18, 35]. In the *prompted self-rationalizing* paradigm, a single LM is instead frozen and prompted to jointly generate the task output and rationale, with the prompt consisting of a few input-output-rationale demonstrations [32]. In the *pipeline-rationalizing* paradigm, a fine-tuned LM first generates the rationale, which is then used as input for a separate fine-tuned reasoning LM to generate the output [14, 24].

However, when considering generalization performance, reliability, and deployment cost, these existing paradigms all have key limitations. Fine-tuned self-rationalizing LMs often perform worse than non-rationalizing LMs, since their parameters are learned using two relatively dissimilar objectives,

2022 Trustworthy and Socially Responsible Machine Learning (TSRML 2022) co-located with NeurIPS 2022.

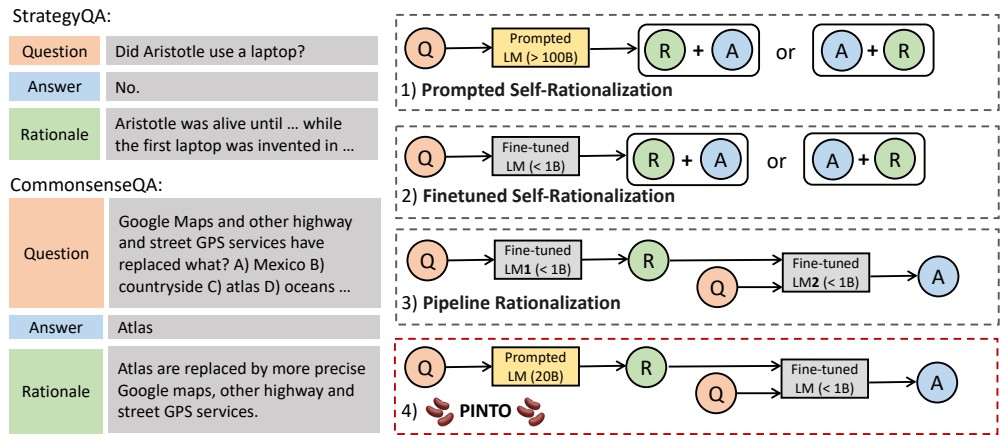

Figure 1: **Rationale-Based Language Reasoning.** (a) Examples of reasoning tasks that require implicit knowledge beyond task inputs. (b) Comparison of existing paradigms for providing free-text rationales along with predictions.

while also requiring expensive rationale annotations [34, 21]. Prompted self-rationalizing LMs yield strong task performance and only need a few rationale demonstrations for the prompt, but are computationally prohibitive since they generally require very large-scale (*i.e.,* over 100B parameters) LMs to work effectively [31, 32]. Besides requiring expensive rationale annotations, pipeline-rationalizing LMs' generated rationale forms a non-differentiable bottleneck between the two modules, which complicates end-to-end training and can hurt task performance [34, 12]. Moreover, none of these paradigms has a mechanism for regularizing the rationale generation to *faithfully* reflect the reasoning process of the LM, without hurting task performance.

In this paper, we propose **P**rompted Rat**I**onalizing with Cou**NT**erfactual Reas**O**ning (🫘 **PINTO**), an LM pipeline that rationalizes via prompt-based learning, then reasons over the task input and rationale via counterfactual regularization. PINTO's *rationalizing module* is a medium-scale (*i.e.,* 20B parameters) LM that contains vast latent knowledge obtained via pretraining [2]. Though prohibitive to fine-tune, it is affordable for prompt-based learning. Given the task input and a minimal input-output demonstration prompt, the rationalizing module uses its internal knowledge to map out a suitable reasoning process for the task input by generating a free-text rationale. The rationalizing module is frozen during fine-tuning, which drastically reduces training costs and prevents it from exploiting spurious shortcuts in the downstream training data. PINTO's *reasoning module* is a small-scale (*i.e.,* under 1B parameters) LM to which knowledge is transferred from the much larger rationalizing module. The reasoning module is fine-tuned to solve the downstream reasoning task by using the generated rationale as context for the task input. Crucially, to help ensure that the rationale dictates the behavior of the reasoning module, the reasoning module is regularized to output less confident predictions when the rationale is perturbed. To simulate shortcut reasoning with rationales, we consider two rationale perturbation strategies: token masking (*i.e.,* rationale is ignored) and token replacement (*i.e.,* rationale is misused).

Across four question answering datasets (CSQA, StrategyQA, OpenBookQA, QASC), we show that PINTO significantly improves the reasoning LM's generalization, yielding higher performance on both in-distribution (ID) and out-of-distribution (OOD) test sets. Also, we find that rationales are leveraged more faithfully by PINTO than by other methods. Furthermore, we show that PINTO's counterfactual regularization improves the reasoning module's robustness to noise in the rationalizing module's generated rationales.

## 2 Rationale-Based Language Reasoning

In this work, we study LMs' ability on language-based reasoning using implicit knowledge. We consider a specific type of multi-choice question answering (QA) tasks where the required knowledge for answering the question is not explicitly provided in the input and needs to be inferred from the LM's parameters [30, 13]: Given a question $q$ and a set of answer choices $A = \{a_i\}$, the model's goal is to predict a plausibility score $\rho(q, a_i)$ for each $(q, a_i)$ pair, so that the predicted answer $\hat{a} = \arg\max_{a_i \in A} \rho(q, a_i)$ matches the correct answer choice $a^* \in A$.

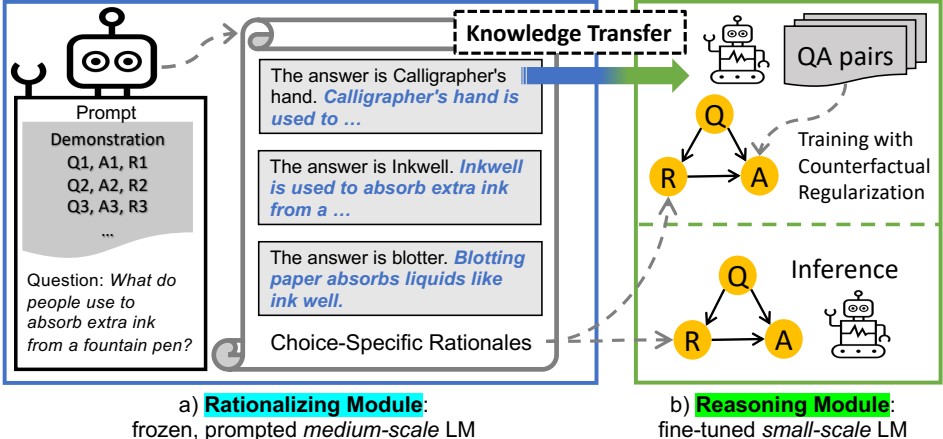

a) **Rationalizing Module**: frozen, prompted *medium-scale* LM

b) **Reasoning Module**: fine-tuned *small-scale* LM

Figure 2: **Overview of PINTO.** (1) A frozen medium-scale LM is prompted to generate choice-specific rationales. (2) A small-scale LM is fine-tuned to reason over the generated rationales. (3) We introduce counterfactual regularization in addition to standard training loss to ensure the rationales are leveraged properly. During inference, the rationalizing LM is prompted with a new question to generate rationales, which are provided to the reasoning module to make a prediction.

Motivated by the LM tendency to exploit reasoning shortcuts for solving tasks [3], we focus on methods that explicitly generate free-text rationales to explain their predictions. Whereas extractive rationales are limited to input token scoring [7, 27, 5], free-text rationales use natural language to describe a reasoning process (*e.g.,* things beyond the task input) [21, 32]. Below, we discuss several paradigms (see also Fig. 1) for rationale-based language reasoning.

**Fine-Tuned Self-Rationalization** In this paradigm, an LM is *fine-tuned* to generate the task output and rationale as a single sequence [21, 17]. If the rationale is generated after the task output, then the rationale is conditioned on the task output, and vice versa. Since the LM parameters are shared across two relatively dissimilar objectives, they often perform worse than non-rationalizing LMs [34, 21]. Notably, this paradigm requires expensive rationale annotations for all training instances.

**Prompted Self-Rationalization** In this paradigm, a pretrained LM is *frozen* and *prompted* to generate the task output and rationale as a single sequence, with the prompt consisting of a few input-output-rationale demonstrations [15, 32]. If the rationale is generated after the task output, then the rationale is conditioned on the task output, and vice versa. This paradigm performs well and it only needs a few rationale annotations for the prompt, but it is computationally prohibitive since it generally requires very large-scale (*i.e.,* over 100B parameters) LMs to work effectively [15, 32].

**Pipeline Rationalization** Here, a fine-tuned rationalizing LM first generates the rationale, which is then used as input for a separate fine-tuned reasoning LM to predict the task output [14, 24]. This paradigm's generated rationale forms a discrete (*i.e.,* non-differentiable) bottleneck between the two modules, which complicates end-to-end training and can hurt task performance [34, 12]. Additionally, the dedicated rationalizing LM requires extra rationale annotation/computation costs.

## 3 PINTO: Faithful Language Reasoning Using Prompt-Generated Rationales

PINTO is a two-stage, rationalize-then-reason pipeline, designed to address the limitations of existing paradigms for rationale-based language reasoning (§2). Like the pipeline rationalization paradigm, PINTO has separate modules for rationalizing and reasoning. However, PINTO's rationalizing module is prompted instead of fine-tuned. As a result, PINTOdoes not suffer from the non-differentiable bottleneck issue and has lower rationale annotation/computation costs.

Following prior works, PINTO is based on choice-specific rationales [14, 12]. First, given $q$ and $A$, the *rationalizing* module generates a set of choice-specific rationales $R = \{r_i\}$, where each $r_i$ explains a reasoning process that supports answer choice $a_i \in A$ (§3.1), as opposed to generating one rationale per question. We opt for this design choice because rationales are often answer-leaking [26], *i.e.,* the rationale itself is already sufficiently predictive of one of the answer choices. If the rationalizing module only generates one rationale per question, then it is forced to make an "early decision", and the reasoning module would only be left to recover the answer from the rationale [14].

Table 1: **Rationalization Prompts.** The format of our prompts for rationalization with a medium-scale LM. The prompt consists of a few examples as demonstration on how to rationalize for a question-choice pair and placeholders for new question and a target choice.

| Task | CommonsenseQA | OpenBookQA |
|---|---|---|
| Prompt | **Q**: What do people use to absorb extra ink from a fountain pen? **Answer Choices**: (a) shirt pocket (b) calligrapher's hand (c) inkwell (d) desk drawer (e) blotter **A**: The answer is blotter. *Blotting paper absorbs liquids like ink well.* | **Q**: How do you reduce pollution? **Answer choices**:(a) igniting fuel and oxidiser (b) transportation technology ... (h) using less resources **A**: The answer is using less resources. *Conserving resources has a positive impact on the environment. Use of resources affects the environment such as pollution.* |

While prior works require expensive rationale annotations to train/prompt the rationalizing module [14, 12], PINTO's rationalizing module is a frozen pretrained LM that uses only a few question-answer demonstrations as a prompt (§3.1). Second, given $q$, $a_i \in A$, and $r_i \in R$, the *reasoning* module outputs plausibility score $\rho(q, a_i, r_i)$ (§3.2). We also design a regularization objective that encourages the reasoning module to properly use the rationales to predict the answer (§3.3).

## 3.1   Rationalizing Module

Prior works mainly rely on human-annotated rationales to teach a model to rationalize [14, 12, 26]. However, such rationale annotations are expensive and frequently low-quality [1, 26, 24], *e.g.,* not providing sufficient knowledge to support a given answer. Meanwhile, rationales automatically generated by pretrained LMs are often preferred over human-annotated rationales [33]. Therefore, for PINTO's rationalizing module, we propose using a pretrained LM to generate rationales via in-context learning, which prompts the frozen LM to retrieve knowledge from its parameters [32].

The prompt consists of a fixed set of question-answer demonstrations that are randomly selected from the training set. Each demonstration consists of a question $q$, answer choices $A$,[1] gold answer $a^* \in A$, and a human-annotated free-text rationale $r^* \in R$ for $a^*$ (Table 1).[2] With this prompt $p$, we use the LM to generate rationales for every instance from the dataset. Specifically, for each $a_i \in A$ of some instance $(q, A)$, the rationalizing LM's input is constructed as $[p, q, A, a_i]$. Then, we use greedy decoding of the LM output to obtain rationale $r_i$ for $a_i$. Note that the LM input does not have any information about the gold answer $a^*$. Our design of the rationalizing module assumes that $r_i$ will be aligned with common sense if and only if $a_i = a^*$, since it should intuitively be difficult to retrieve correct commonsense knowledge that supports an incorrect answer choice. The reasoning module then predicts the correct answer by reasoning over the rationales for each answer choice.

## 3.2   Reasoning Module

Given a question $q$, the answer choices $A$, answer candidate $a_i \in A$, and rationale $r_i$, the reasoning module learns to output plausibility score $\rho_i = \rho(q, A, a_i, r_i)$. Following prior works, we use a text-to-text Transformer LM as the backbone of our reasoning module [34, 12]. For each $a_i$, the reasoning module's input is defined as the token sequence $s = [q \oplus a_1 \oplus ... \oplus a_{|A|} \oplus r_i]$, where $\oplus$ denotes concatenation. Meanwhile, the reasoning module's output is obtained by sequentially teacher-forcing $a_i$'s tokens $t_i = [t_1^i, t_2^i, ..., t_{|a_i|}^i]$ into the decoder, rather than via greedy decoding. This way, we can compute the reasoning module's output token probabilities for arbitrary answer choices $a_i$. Following [25], we compute $a_i$'s plausibility score $\rho_i$ by aggregating $t_i$'s token probabilities as:

$$\rho_i = \frac{1}{|a_i|} \sum_{j=1}^{|a_i|} P(t_j^i \,|\, t_{j-1}^i, ..., t_2^i, t_1^i, q, a_i, r_i).$$

Next, we use the softmax function to normalize $\rho_i$ as probability $P(a_i \,|\, q, A, R) = e^{\rho_i} / \sum_{j=1}^{|A|} e^{\rho_j}$. During **inference**, the rationalizing module firstly generate $R = \{r_i\}$ given a new question and answer choices, and then the predicted answer choice is computed by the reasoning module as $\hat{a} = \arg\max_{a_i \in A} P(a_i \,|\, q, A, R)$.

---

[1]We include the answer choices $A$ in the prompt so that the LM is aware of all the available choices and thus could generate a rationale that is more distinctive.

[2]As opposed to full human annotation, we only need a few (usually <8) examples for one dataset.

### 3.3 Training

For a multi-choice QA, the standard training objective is to maximize the likelihood of the correct answer choice using cross-entropy loss, computed as:

$$\mathcal{L}_{\text{std}} = -\sum_{a_i \in A} Q(a_i \,|\, q, A) \log P(a_i \,|\, q, A, R), \qquad (1)$$

where $Q(a_i \,|\, q, A)$ is 1 if $a_i = a^*$ and 0 otherwise. Let $Q(A \,|\, q, A)$ be the one-hot target distribution over all $a_i \in A$. There can be spurious correlations between $q$ and $A$ [3], so the reasoning module may take undesirable shortcuts instead of properly using the rationale to predict the answer [11, 19]. In this case, the rationales would be unfaithful in explaining the model's behavior and useless for model debugging.

To address this, we introduce a counterfactual regularization objective in which the reasoning module is regularized to output less confident predictions when the rationale is not leveraged properly (*i.e.,* shortcuts are used). This is implemented using label smoothing [28], which softens the target distribution $Q(A \,|\, q, A)$ by linearly combining it with a noisy distribution $U(A \,|\, q, A)$, often set as the uniform distribution. Therefore, given tunable label smoothing factor $0 < \epsilon < 1$, we compute the label-smoothed target distribution as: $Q'(A \,|\, q, A) = (1 - \epsilon) \, Q(A \,|\, q, A) + \epsilon \, U(A \,|\, q, A)$.

In order to simulate shortcut reasoning, we consider the two strategies for perturbing the generated rationales $r_i$. **Token Masking** addresses the case where the reasoning module ignores the rationale and instead exploits spurious cues in the rest of the input. To simulate this, we mask the rationales

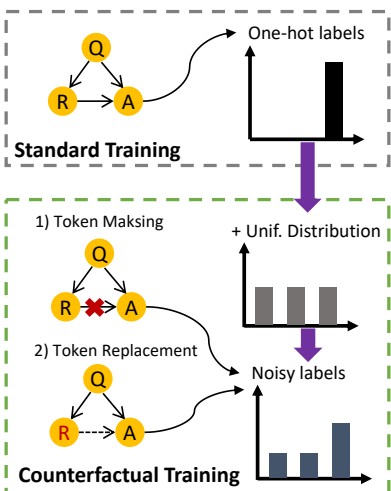

Figure 3: **Standard Training vs. Counterfactual Training.** For counterfactual regularization, we train the reasoning module with noisy labels when the rationale tokens are either masked or replaced.

from the input. Recall that the backbone of the reasoning module is a Transformer LM, which uses a self-attention mechanism to aggregate information across tokens. Hence, we implement rationale masking by zeroing the attention mask for rationale tokens.[3] **Token Replacement** addresses the scenario in which the reasoning module misunderstands the rationales. It randomly replaces $k\%$ of the rationale tokens with other tokens randomly sampled from the entire vocabulary.

At each fine-tuning step, we randomly select one of the strategies for obtaining perturbed rationales $R' = \{r'_i\}$, which helps keep the LM from overfitting to any particular strategy. Then, the counterfactual regularization loss is computed as:

$$\mathcal{L}_{\text{c-reg}} = -\sum_{a_i \in A} Q'(a_i \,|\, q, A) \log P(a_i \,|\, q, A, R'). \qquad (2)$$

This counterfactual regularization teaches the reasoning module to be less confident when the rationales are either absent or problematic, so that it can learn to make sounder use of the rationales.

## 4 Experimental Setup

**Datasets** We experiment with several CSR benchmarks. (1) CommonsenseQA [29], (2) StrategyQA [10], (3) OpenBookQA [20], and (4) QASC [13]. Since the gold labels for the testing sets of these datasets are not publicly available, we treat the official development set as our test set, and separate the training data into our own training set and development set.

**Baselines** (1) *Prompted Self-Rationalization* is a GPT-neox LM which learns from a few examples in the prompt to firstly generate a few short sentences as the rationale and then predict the answer. We use the chain-of-thought type of prompt as in [32]. (2) *Without Rationales* is a T5-based model fine-tuned with the task dataset without using any rationales as additional input. (3) *Standard Training* adopts the same rationalizing and reasoning pipeline as our method, but the reasoning module is not

---

[3]We do not choose to replace the tokens in a rationale with special mask tokens since the LM is already pretrained to recover the mask tokens, and we want to ensure that this ability is completely deprived.

Table 2: **ID Results.** Task performance (accuracy) and faithfulness (LAS) of the compared methods on the testing datasets. The rationalizing module is GPT-neox (20B) while the reasoning module for the fine-tuning methods is T5-based. The Prompted Self-Rationalization using GPT-3 is reported to achieve 73.50 and 66.53 in accuracy on CSQA and StrategyQA respectively [32]. We bold the results that exceed the second best with statistical significance (p-value< 0.05).

| Method | CSQA | | StrategyQA | | OpenBookQA | | QASC | |
|---|---|---|---|---|---|---|---|---|
| | Acc.↑ | LAS↑ | Acc.↑ | LAS↑ | Acc.↑ | LAS↑ | Acc.↑ | LAS↑ |
| Prompted Self-Rational. | 38.41 | 11.66 | 55.31 | 1.09 | 33.80 | 14.67 | 32.61 | 32.01 |
| w/o Rationales | 58.68 | - | 58.12 | - | 55.85 | - | 35.58 | - |
| Standard Training | 59.48 | 18.75 | 57.11 | 1.50 | 56.65 | 17.03 | 37.50 | 37.91 |
| Dropout Context | 59.64 | 20.40 | 51.45 | 0.62 | 57.55 | **18.76** | 35.37 | 37.54 |
| PINTO | **61.67** | **24.22** | **60.87** | **3.35** | 58.85 | 18.02 | 37.82 | **38.98** |
| - Token Masking Only | 60.46 | 17.44 | 59.12 | 1.74 | 58.35 | 13.06 | 37.39 | 34.06 |
| - Token Replacement Only | 60.38 | 22.54 | 58.72 | 2.11 | 58.10 | 18.01 | 37.47 | 34.61 |

Table 3: **OOD Results.** Performance (accuracy) of the compared methods, which are firstly trained on a source dataset and then directly predict on a target dataset (denoted as $source \rightarrow target$).

| Method | CS→OB | CS→QASC | OB→CS | QASC→CS | QASC→OB |
|---|---|---|---|---|---|
| w/o Rationales | 32.05 | 39.17 | 24.87 | 45.74 | 34.90 |
| Standard Training | 31.05 | 40.04 | 25.37 | 47.71 | 34.50 |
| Dropout Context | 32.30 | 38.85 | 23.01 | 44.27 | 32.90 |
| PINTO | **34.90** | **42.25** | **27.66** | **48.03** | **35.75** |

fine-tuned with the counterfactual training loss. (4) *Dropout Context* is the same as the Standard Training baseline except that during the fine-tuning of the reasoning module, the question is randomly dropped out from the input, which is a strategy adopted in prior work [12] to encourage the reasoning module to make good use of the input rationales.

Implementaion details of our method can be found in the Appendix (A.1). We also consider two PINTO variants as baselines: *Token Masking Only* and *Token Replacement Only*. These only use token masking or token replacement for perturbing rationale tokens, respectively.

**Evaluation Metrics** To evaluate *task performance*, we measure accuracy and consider both ID and OOD evaluation sets in our experiments. To evaluate the *faithfulness* of the generated rationale to the model's prediction, we adopt LAS metric [12], which measures how well rationales help a simulator predict a model's output. Following [12], we use a fine-tuned T5-base for the simulator.

## 5 Experiments

### 5.1 Main Results

**Performance on ID data** Table 2 shows the task performance in accuracy of all the compared methods on the four CSR datasets we consider, from which we make two observations. First, the Prompted Self-Rationalization baseline with GPT-neox (20B) generally does not outperform the fine-tuning methods while the GPT-3 version is reported to achieve 73.50 and 66.53 in accuracy on CSQA and StrategyQA respectively [32]. This validates that Prompted Self-Rationalization requires very large LMs to work effectively [31]. Second, simply augmenting the reasoning module with rationales (as in Standard Training) does not always lead to better results compared with the Without Rationales baseline since the rationales may not be properly utilized. The Dropout Context baseline helps to address this issue in some, but not all cases, while PINTO consistently yields the best accuracy in most of the cases.

**Generalizability to OOD data** To demonstrate the generalizability brought by faithful reasoning over rationales, we further investigate the performance of our method on out-of-distribution (OOD) data. The intuition is that by utilizing rationales faithfully rather than fitting only the in-distribution training data, our model achieves better OOD generalization without any fine-tuning. Table 3 shows the OOD performance of all the fine-tuning methods. We conclude that rationales are helpful in improving the generalizability of the model to a dataset unseen during fine-tuning. Among all the

methods utilizing rationales, our method yields the best OOD performance, which confirms the benefit of faithful reasoning.

**Rationale-Label Association** Table 2 also reports the faithfulness of all the methods involving rationalization measured by LAS. We observe that PINTO achieves a much higher score compared with the baselines except on OpenBookQA. This demonstrates that with counterfactual regularization, the reasoning module can make predictions more faithfully with regard to the rationales.

## 5.2 Performance Analysis

**Can we refine the reasoning behavior via rationales?** One important application of faithful reasoning is that rationales provide a way to refine the behavior of a model, i.e., we can correct reasoning mistakes by providing a better rationale. To verify this, we make use of ECQA [1] which augments CSQA with human-annotated rationales. We directly provide the human-annotated rationales to the fine-tuned reasoning modules to obtain its oracle results, shown in Figure 4. We see that human-annotated rationales generally lead to performance gain for all fine-tuning methods whereof the gain of our method is the largest. This again showcases the merits of ensuring the faithful reasoning on rationales in refining a system.

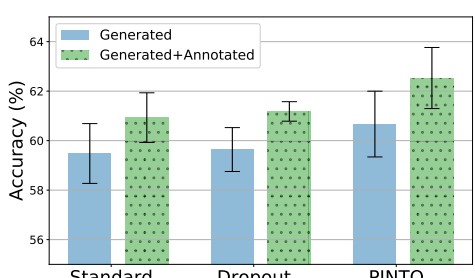

Figure 4: **Rationale Quality Analysis.** Accuracy of models with both generated and annotated rationales vs. models using only generated rationales on CSQA.

**Is our method more robust to perturbed rationales?** One potential risk with the rationalize-then-predict approach is error propagation where the model may be misled by less optimal rationales. However, since the label smoothing loss used for counterfactual regularization still teaches the reasoning module to output a reasonable distribution with the answer having the highest probability, the ability to reason robustly is thus preserved in our method. To

Table 4: **Robustness to Noisy Rationales.** Robustness (accuracy) of the compared methods. We use perturbed rationales during inference as a stress test.

| Model / Rationale | CSQA | | OBQA | |
|---|---|---|---|---|
| | Original | Perturbed | Original | Perturbed |
| Standard Training | 59.48 | 58.60 | 56.65 | 56.30 |
| Dropout Context | 59.64 | 57.58 | 57.55 | 57.00 |
| PINTO | **61.67** | **59.05** | **58.85** | **57.30** |

verify this, we conduct a stress test. We choose CSQA and OpenBookQA and replace each question in the testing set with a randomly sampled question but still keep the original answer choices. We then prompt our rationalizing module with the replaced question and the original choices to obtain a set of perturbed rationales. We finally provide the perturbed rationales to the reasoning module. Table 4 shows the results. We can observe that among all the methods, PINTO still predicts robustly without much performance drop while baselines are affected more severely. Note that the robustness is an additional advantage brought by counterfactual regularization and is not contradictory to our goal of ensuring faithfulness, since normally we do not use the perturbed rationales for inference.

## 6 Conclusion

This paper presents PINTO, an LM pipeline that rationalizes with prompt-based learning and reasons via counterfactual regularization. Through prompting, we remove the need for expensive human annotation and leverage the massive knowledge encoded in a medium-sized LM to perform rationalization. With counterfactual regularization in addition to standard training objective, our reasoning module learns to reason over the generated rationales more faithfully. Experiments show that our method outperforms baselines on both in-distribution and out-of-distribution datasets in accuracy, while providing higher faithfulness. Our analysis also shows that we can further improve task performance with a more faithful reasoning module and refined rationales.

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

# A Appendix

## A.1 Implementation Details

For the rationalizing module, we use GPT-neox [2], a pretrained, autoregressive LM with 20B parameters. We manually annotate 7 examples to set up the prompt for each task dataset. For the reasoning module, we adopt T5-base [22] with only 220 million parameters, which is around two orders of magnitude smaller than the rationalizing module. During fine-tuning, the standard training loss (Eq. 1) and our counterfactual training loss (Eq. 2) are directly combined as the overall training loss. For perturbing rationales, we randomly choose the token masking or token replacement strategy with a equal chance in each training batch. The replacing rate for token replacement is empirically set to 30%. We run all the experiments on the compared methods 4 times using a fixed set of random seeds and report the average results.

## A.2 Case Study

Table 5: **Case Study.** Generated rationales and model predictions from CSQA and StrategyQA.

| Questions and Choice-specific Rationales | Standard | PINTO |
|---|---|---|
| Q: If you have leftover cake, where would you put it?
(a) quandry: *Quandry is a place where you put things that you don't know ...*
(b) refrigerator: *Refrigerator is used to store food.*
(c) oven: *Oven is used to bake cakes.* | oven ✗ | refrigerator ✓ |
| Q: Has every astronaut survived their space journey?
yes: *Astronauts have died in space. For example, the Challenger astronauts died in the Challenger space shuttle explosion.*
no: *The first astronaut to die was Yuri Gagarin. Gagarin died in a plane crash.* | yes ✗ | no ✓ |

We provide concrete examples in Table 5 to showcase how our prompted LM rationalizes for correct and incorrect choices and how PINTO reasons more faithfully compared with the Standard baseline. In the question (second row) from CSQA, we can see that for incorrect choices, the generated rationales do not support them to be the answer while the one for the correct choice *refrigerator* does. In the question (third row) from StrategyQA, the rationale for the correct choice *yes* is sound and reasonable while the rationale for the incorrect choice *no* is factually correct but does not answer the question directly (*died in a plane crash* vs. *died in the space journey*). For both questions, PINTO properly leverages the rationales and make the correct predictions while the Standard baseline fails.

