# OpenReview forum: "PINTO: Faithful Language Reasoning Using Prompt-Generated Rationales"
_NeurIPS.cc/2022/Workshop/TSRML — TSRML2022_

### Official Review · Reviewer_a34F · 2022-10-21
**Review for PINTO**

**Overall Recommendation:** The paper is well-organized and I wou…
**Overall Rating:** 7

**Summary:**

This paper proposes PINTO, a language model pipeline that rationalizes through prompt-based learning, and learns to faithfully reason over rationales via counterfactual regularization. PINTO first generates rationales through a frozen LM and then uses the rationals as contexts to fine-tune a reasoning LM. The reasoning LM is regularized to generate less confident predictions when the rationals are perturbed.

**Strengths:**

 - The experiments are extensive with several CSR benchmarks, showing that the proposed PINTO outperforms baselines by a large margin in most of the experiments.
 - The ablation studies show that both the rationales and the regularization help improve the performance.


**Weaknesses:**

 - It would be interesting to conduct another ablation study by utilizing LM with different model sizes.

**Review Confidence:**

4: The reviewer is confident but not absolutely certain that the evaluation is correct

---

### Official Review · Reviewer_XguF · 2022-10-22
**An interesting use of prompting; I'd like the technical sections to be more precise**

**Overall Rating:** 7

**Summary:**

The paper seeks to improve on prior work on providing natural language reasoning. The authors seek to create a model that provides a rationale for why it chose a certain response to a multiple choice answer through a two stage process where the first stage generates a candidate rationale and the second stage chooses the most plausible rationale for the query and the set of candidate answers.  The primary contribution seems to be a pipeline that uses prompting to make what prior work has expressed as a non-differentiable 2 stage method, differentiable. The authors then show results that confirm that the authors' approach improves over prior work in this area.

**Strengths:**

Figures 1 and 2 are extremely good for providing a high level idea of what this approach is doing. Moreover, I think that prompting is a very good idea to improve on the lack of differentiability with the Pipeline Rationalization.

Your results seem to show that PINTO works well and improves over prior work on rationalization. I particularly like the additional work that you all did to show that the accuracy of the model can be improved by improving the annotations over time.


**Weaknesses:**

Your reasoning model was somewhat confusing for me. The rationalizing module provides a token sequence, s = [ q + a_{1} + ... + a_{|A|} + r_{i} ], and then you pass a list of tokens t_{i} = [t_{1}^{i}, ... t_{|a_{i}|}^{i}] for every a_{i} into the decoder, but I'm not clear on what these tokens are. If I understand it correctly, we have a tokenizer, eg. a byte pair encoding, that takes the outputs of the rationalizing module, and then converts that into the token list t_{i}. But, we already also keep the non-tokenized rationale r_{i}. The plausibility of a rationale is determined by the joint distribution over the t_{i} tokens, conditioned on the predicted response a_{i|, the query, q, and the set of tokens r_{i}.

I don't understand why we need to keep r_{i} if we have the tokens that make up r_{i}. Up to this point, I thought I understood what a_{i} is, but when you express | a_{i} |, I lose track of what these things are. Is a_{i} also a set of tokens and | a_{i} | is length? If so why do we express r_{i} in terms of its tokens t_{i} separately? Moreover P( t_{i} | r_{i} ) wouldn't make sense if t_{i} are just the tokenization of r_{i}. I'm willing to assume that I am missing something here, as your results and figures seem to show that this approach works well, but sections 3.2 and 3.3 need to be precise and clear.

(As an aside, not everything needs an acronym. If you have to use the "O" in "Reasoning" to make your acronym work, it feels a bit too forced)

**Overall Recommendation:**

I think that this is a good paper, that is worth being accepted to the workshop. The contribution seems sound, however, I feel that the authors should clarify and make more precise sections 3.2 and 3.3.

**Review Confidence:**

3: The reviewer is fairly confident that the evaluation is correct

---

### Decision · Program_Chairs · 2022-10-23

**Decision:**

Accept

**Comment:**

Following the unanimous recommendations from reviewers, the submission is accepted.